# New Antimicrobials Based on the Adarotene Scaffold with Activity against Multi-Drug Resistant *Staphylococcus aureus* and Vancomycin-Resistant *Enterococcus*

**DOI:** 10.3390/antibiotics10020126

**Published:** 2021-01-28

**Authors:** Salvatore Princiotto, Stefania Mazzini, Loana Musso, Fabio Arena, Sabrina Dallavalle, Claudio Pisano

**Affiliations:** 1Department of Food, Environmental and Nutritional Sciences (DeFENS), Università degli Studi di Milano, Via Celoria 2, 20133 Milan, Italy; salvatore.princiotto@unimi.it (S.P.); stefania.mazzini@unimi.it (S.M.); loana.musso@unimi.it (L.M.); 2Centro di Ricerche Biomediche, Department of Clinical and Experimental Medicine, University of Foggia, Via Luigi Pinto, 71122 Foggia, Italy; 3IRCCS Don Carlo Gnocchi Foundation, 50143 Florence, Italy; 4Preclinical Research & Biotech Development, Special Product’s Line (Anagni), 03012 Frosinone, Italy; c.pisano@specialspa.it

**Keywords:** adarotene, antimicrobials, antimicrobial resistance, gram-positive, bactericidal activity

## Abstract

The global increase in infections by multi-drug resistant (MDR) pathogens is severely impacting our ability to successfully treat common infections. Herein, we report the antibacterial activity against *S. aureus* and *E. faecalis* (including some MDR strains) of a panel of adarotene-related synthetic retinoids. In many cases, these compounds showed, together with favorable MICs, a detectable bactericidal effect. We found that the pattern of substitution on adarotene could be modulated to obtain selectivity for antibacterial over the known anticancer activity of these compounds. NMR experiments allowed us to define the interaction between adarotene and a model of microorganism membrane. Biological assessment confirmed that the scaffold of adarotene is promising for further developments of non-toxic antimicrobials active on MDR strains.

## 1. Introduction

Bacterial resistance to antimicrobial drugs is becoming one of the major threats to human health. The increasing occurrence of infections by multi-drug resistant (MDR) pathogens is associated with high mortality and morbidity [1] and the lack of new antimicrobial agents for treatment of these infections has led to serious concerns.

Bacterial resistance can occur in multiple ways, including the modification or overexpression of the antibiotics target, the decrease in the intracellular antibiotic concentration, either by utilization of efflux systems actively transporting the drug out of the cell or by mechanisms that reduce their influx, and the expression of enzymes able to inactivate the antibiotic [2,3]. Though there are no extensive data that globally define the phenomenon of antimicrobial resistance (AMR), the latest estimates show that it is the cause of over 700,000 deaths per year worldwide [4,5]. In accordance with the data provided by the European Commission [6], which investigated the epidemiology of the phenomenon starting from 2009, around 25,000 patients die each year in the EU for infections caused by MDR bacteria [7]. It has been estimated that if the phenomenon is not curbed by 2050 there will be 10 million people worldwide who will die from multiple antibiotic resistance [8]. It is therefore clear that the need for novel antimicrobial drugs counteracting the development of bacterial resistance is imperative.

*Staphylococcus aureus* has emerged as one of the most threatening microorganisms. *S. aureus* is a commensal of upper respiratory tract and skin of a significant proportion of the general human population and, in certain circumstances, it is also able to cause virtually all kind of pyogenic infections [9]. *S. aureus* resistant to methicillin (MRSA), in particular, is among the most challenging MDR pathogens [10].

Furthermore, *Enterococcus* spp. resistant to vancomycin, another MDR Gram-positive pathogen, is reported on the World Health Organization’s 2017 list of high-priority pathogens for which new treatments are urgently needed [11]. 

In a recent publication, Kim and colleagues [12] reported that the retinoid related molecules (RRM) CD437 and CD1530 (Figure 1) exhibit activity against methicillin-resistant *S. aureus* (MRSA) strain MW2. The compounds showed high killing rates by disrupting lipid bilayers, synergism with gentamicin, and a low probability of resistance selection, in a *C. elegans*-MRSA killing assay. These relevant findings open a new avenue of investigation.

During the last years we have performed extensive SAR studies to investigate the antitumor activity of RRM [13,14,15,16]. Having synthesized a large number of compounds containing the adarotene scaffold (Figure 1), we selected some representative derivatives for antimicrobial activity screening. However, the major obstacle in developing retinoids as antimicrobials is their cytotoxicity, most of them being endowed with antitumor activity. For this reason, in our initial screening, we selected those compounds that showed lower antitumor activity in our previous studies (IC_50_ > 1 µM), to focus the investigation on analogues with a low-toxicity profile.

The molecules considered in this study include compounds with diverse substituents on the aromatic scaffold to explore the role of key portions of adarotene: the adamantyl moiety (compound **1**), the carboxylic acid group (compounds **4**, **5**, **6**) and the α,β-unsaturated system (compounds **7**, **8**, **9**, **10**). Additionally, we tested analogues featuring substituents on rings A and B of the biphenyl skeleton (compounds **11**–**15**, ring A, **16**–**20**, ring B). Compounds **11** and **12**, endowed with potent cytotoxic activity (IC_50_ 0.2–0.5 µM), were selected to evaluate a possible correlation between cytotoxic and antimicrobial activity. We also synthesized new compounds (**2**, **9**, **14**) to widen the coverage of the chemical space and to perform preliminary structure–activity relationship (SAR) studies (Figure 2).

## 2. Results and Discussion

### 2.1. Chemistry

Compounds **1**, **3**–**8**, **10**–**13**, **15**–**20** were prepared as previously described [13,14,15,16,17].

Adarotene analogues **2**, **9** and **14** were synthesized as reported in Scheme 1. Alkylation of compound **21** [13] with 4-bromobutyl acetate in presence of K_2_CO_3_ in DMF, followed by basic hydrolysis gave compound **2** in 37% yield. 

Aldehyde **22** [13] was used as a starting material for the introduction of a cyano group on the double bond. Condensation with methyl cyanoacetate and hydrolysis of the methyl ester group gave compound **9**. 

The reaction of 2-chloroacetylamine **23** [14] with phenyl isocyanate gave the ureido analogue **14**. 

### 2.2. Antimicrobial Activity

The compounds were initially tested with a methicillin-resistant *S. aureus* and an *E. faecalis* clinical isolates (details in the experimental section), two MDR strains according to definition by Magiorakos et al. [18]. The activity of studied compounds was assessed by broth micro-dilution method. Minimal Inhibitory Concentration (MIC), and Minimal Bactericidal Concentration (MBC) were determined and possible bacteriostatic or bactericidal activity was also investigated. Adjunctively, the compounds were tested against *Escherichia coli*, *Acinetobacter baumannii* and *Pseudomonas aeruginosa* strains, however they all showed absent or low activity (MIC >128 µg/mL, data not shown). Antimicrobial activity of studied compounds is shown in Table 1.

MICs of tested compounds ranged widely from 1 – >256 µg/mL. However, the majority of compounds showed detectable antimicrobial activity in the range MIC: 1–128 µg/mL against both *S. aureus* and *E. faecalis* strains. As the molecular weights of the compounds listed in Figure 2 are slightly different each other, the MIC and MBC are reported in µM concentrations, as well (Table 1). The data obtained confirmed the trend of activity reported in µg/mL. 

To shed light on the molecular determinants of the activity we performed preliminary SAR studies by separately analysing the effect of modifications on the key portions of adarotene. The replacement of the liphophilic bulky adamantyl group with a *t*-butyl group (compound **1**) was deleterious, causing a decrease in activity. This is consistent with the results very recently obtained by Cheng and colleagues on the CD437 scaffold, which proved that the adamantyl substitution is optimal for the antimicrobial activity [19].

Introduction of substituents on the phenolic OH also caused a decrease in activity. Compound **2**, with an OH-ending alkyl chain linked to the phenolic group was the worst analogue of the series, with a MIC > 256 µg/mL for both *S. aureus* and *E. faecalis* strains. Additionally, the introduction of a lipid-mimicking unsaturated alkyl chain (compound **3**) caused an increase in MICs (32 µg/mL for *S. aureus*, 128 µg/mL for *E. faecalis*), confirming that the phenolic hydroxy group plays an essential role for activity. The substitution of the carboxylic acid moiety with a cyano, a hydroxamic acid or a *p*-OHphenylamide was also deleterious, giving compounds (**4**, **5**, **6** respectively) with MIC >= 32 µg/mL. 

Conversely, introduction of substituents on the double bond increased the activity. Compounds **7**, **8**, **9** had a MIC 2–4 µg/mL for *S. aureus* and 1–4 µg/mL for *E. faecalis* and a bactericidal activity in all except one case. 

The saturation of the chain to give a more flexible system gave a compound with reduced activity (Compound **10**, MICs = 16 µg/mL for both microorganisms). 

Substitution on ring A (compounds **11**–**15**) maintained the activity even in the presence of quite bulky groups. The ethylenedioxy derivative **12** was also endowed with significant activity (MIC 4 µg/mL for *S. aureus* and 2 µg/mL for *E. faecalis*). However, it should be stressed that compounds **11** and **12** had also significant cytotoxic activity. Compound **15** with a small polar formyl group on ring A showed a drop in activity (MIC 64 µg/mL for *S. aureus* and *E. faecalis*).

The same trend was observed when substituents were introduced on ring B. Featuring ring B with lipophilic groups gave potent antimicrobial compounds with bactericidal effect (**16** and **17**, MIC = 4 and 2–8 µg/mL for *S. aureus* and *E. faecalis*, respectively). Conversely, the introduction of hydrophilic acidic or basic groups reduced the activity (compounds **18**–**19**, MIC in the range 32–128 µg/mL for both microorganisms). The introduction of a formyl group on ring B gave, in this case, a compound that maintained a good activity (compound **20** vs compound **15**).

The data demonstrated that the antimicrobial activity can be modulated by modification of both the backbone and substituents and is not correlated to the cytotoxic activity, most likely because of the different mechanism of action of the compounds on microorganisms and human cell lines. Worth of note is that the introduction of bulky substituents on ring B caused a drop in cytotoxic activity (compounds **16**–**19**, IC_50_ > 10 µM), whereas the antimicrobial activity was maintained. In particular, compounds **16** and **17** were among the most potent adarotene-derivatives. 

To expand the previously obtained results compounds **16** and **17** were also tested on wider a selection of strains: two other MDR *S. aureus* and *E. faecalis* clinical isolates, the *S. aureus* ATCC 25923 and the *E. faecalis* ATCC 51299 reference strains. Compound **2**, which was endowed with low antimicrobial activity, was tested as well, to confirm its low efficacy. Adarotene was used as control compound (Table 2). The data obtained on the additional strains confirmed the trend of activity reported in Table 1.

Kim and colleagues [12] reported that the antimicrobial activity of CD-437 derived retinoids was due to their ability to penetrate and embed in lipid bilayers.

The membrane-disruptive action of retinoids could be partially attributed to the interaction with the membrane components, which consequently causes conformational changes and an increase in fluidity, eventually resulting in membrane dysfunction and cell death.

From our results, it also emerged that for the series of adarotene derivatives, two polar groups and a quite rigid and compact skeleton could be important for membrane attachment and penetration. The effect on the bilayer may be ascribed by the penetration of the compounds (LogP of the most active compounds in the range 5.31–6.94) into the lipid molecules through hydrocarbon attraction at the lipid tail region and the hydrogen bonds between the phenolic hydroxyl group and carboxylic acid and the lipid headgroup. The lower activity of compounds **2** and **4** (LogP 5.97 and 6.15, respectively) can be explained by the absence of the crucial moieties to attack the microorganism membrane [12].

### 2.3. NMR Investigation

Kim and colleagues studied the interactions between CD437-derived retinoids and the lipid bilayer by molecular dynamics simulations [12].

To obtain an experimental evidence of the positioning of the RRM into the membranes, which could be useful for a rational design of adarotene analogues, we investigated the attachment of adarotene to a model of microorganism membrane by NMR spectroscopy.

One of the most common models of biological membranes for NMR studies are SDS (sodium dodecylsulphate) micelles, showing a head polar group (the sulphate moiety) that mimics the surface of the membranes. Moreover, the SDS micelles have a larger correlation time with respect to the NMR time-scale and their small size allows a good spectral resolution [20,21].

The results of the study are reported in Figure 3. For the SDS molecule, four peaks can be observed in ^1^H NMR spectrum: the methylene group close to the hydrophilic head of SDS (CH_2_-12, 4.05 ppm), the neighboring methylene group (CH_2_-11, 1.69 ppm), the terminal methyl group (CH_3_-1, 0.89 ppm), and the nine remaining methylenes (1.4 ppm).

The ^1^H-NMR spectrum of adarotene shows two signals in the range of 7.7–7.4 ppm attributed to aromatic and olefinic Ha protons, a signal at 6.95 ppm assigned to the aromatic proton in *ortho* position with respect to the hydroxy group on the ring A (H *ortho*) and the signal of olefinic proton in alpha to the carboxylic group (Hb, 6.49 ppm).

In addition, three signals attributed to the adamantyl moiety have been observed at high field region (2.25–1.82 ppm) (Figure 3).

2D NOESY experiments were initially performed to study the interactions of adarotene with the cell membrane mimetic. The observation of NOE cross-peak between two protons at a sufficiently short mixing time allows to deduce that the protons are close within a distance of 5 Å. However, spin diffusion (indirect magnetization transfer) can occur at long mixing times and this could cause imprecise conclusions on the spatial interaction between protons (data not shown). In order to avoid this problem, ROESY (Rotating frame Overhauser effect spectroscopy) spectra were performed at different mixing times ranging from 250 to 400 ms (Figure 4), confirming that the observed cross-peaks are due to primary NOE interactions.

Despite the overlap of most aromatic protons of adarotene, NMR experiments gave a certain number of unambiguous intra and intermolecular NOE contacts. Besides the obvious intramolecular interactions, a significant intramolecular NOE between Hb and the adamantyl moiety was detected, suggesting the proximity of these different parts of the molecule (Figure 4). Besides all the intramolecular NOEs, some intermolecular NOEs between SDS and adarotene were identified in the 2D ROESY spectra. The position of adarotene with respect to the mimetic membrane was defined by unequivocal NOEs cross peaks involving the isolated olefinic proton (Hb), the adamantyl moiety and the aromatic proton (H ortho) on ring A. Specifically, H ortho interacts with methylene protons 11 and 12 of the polar head of SDS. The adamantyl group shows a contact with CH_2_-11 of SDS as well. In addition, the above cited protons together with all the aromatic protons of adarotene show contacts with the methylene and methyl groups of the lipophilic chain of SDS. Interestingly, no NOEs interactions were found between Hb and CH_3_ (1) of SDS and between Hb and the chain. This means that adarotene is not located on the polar surface of the micelles, but can penetrate the model membrane and forms molecular complexes with SDS.

Overall, the results allowed to define the position of adarotene in SDS micelles as internally oriented with respect to the SDS molecules (Figure 5). These findings provide experimental support and complement the previously reported simulations by Kim and colleagues [12], offering insights into the interaction between RRM and surfaces of membranes, which could help the design of new bioactive adarotene derivatives.

## 3. Materials and Methods

### 3.1. Chemistry. General Information

All reagents and solvents were reagent grade or were purified by standard methods before use. Melting points were determined in open capillaries and are uncorrected. Solvents were routinely distilled prior to use; dry methylene chloride was obtained by distillation from phosphorus pentoxide. All reactions requiring anhydrous conditions were performed under a positive nitrogen flow, and all glassware were oven dried. Isolation and purification of the compounds were performed by flash column chromatography on silica gel 60 (230–400 mesh). Analytical and preparative thin-layer chromatography (TLC) were conducted on TLC plates (silica gel 60 F_254_, aluminium foil) and spots were visualized by UV light and ⁄ or by means of dyeing reagents.

^1^H spectra and ^13^C of the new compounds were recorded on Bruker AMX 300 MHz spectrometer. Chemical shifts (δ values) and coupling constants (J values) are given in ppm and Hz, respectively. Analyses indicated by the symbols of the elements or functions were within ±0.4% of the theoretical values.

Compounds **1**, **3**–**8**, **10**–**13**, **15**–**23** were prepared as previously described [13,14,15,16,17].

*3-[3′-adamantan-1-yl-4′-(hydroxybutoxy)-biphenyl-4-yl]-acrylic acid* (**2**). To a mixture of 3-(3’-adamantan-1-yl-4’-hydroxy-biphenyl-4-yl)-acrylic acid methyl ester (100 mg, 0.26 mmol) in DMF (4.50 mL), 4-bromobutyl acetate (67.4 mg, 0.34 mmol) and K_2_CO_3_ (102 mg, 0.74 mmol) were added and the reaction was heated 2 h at 80 °C. Then, K_2_CO_3_ was filtered, DMF evaporated and the residue was diluted with ethyl acetate and washed with 1M HCl, water and brine. The organic phase was dried over anhydrous Na_2_SO_4_ and the solvent evaporated. Purification by flash chromatography (petroleum ether/ethyl acetate 80: 20) gave 98 mg (75 %) of 3-[4′-(4-acetoxybutoxy)-3′-adamantan-1-yl-biphenyl-4-yl]-acrylic acid methyl ester as white solid, m.p. 157 °C, *R*_f_ (petroleum ether: ethyl acetate 80:20) 0.50. ^1^H-NMR (300 MHz, DMSO-*d*_6_) δ: 7.76–7.60 (4H, m); 7.56 (1H, d, J =16.5 Hz); 7.48 (1H, dd, J = 8.5, 2.1 Hz); 7.43 (1H, d, J = 2.1 Hz); 6.95 (1H, d, J = 8.5 Hz); 6.51 (1H, d, J = 16.5 Hz); 4.84 (2H, s); 3.73 (3H, s); 4H missing due to the overlap with signal solvent; 2.20–1.98 (9H, m); 1.84–1.65 (6H, m); 1.47 (7H, s).

A stirred suspension of 3-[4′-(4-acetoxybutoxy)-3′-adamantan-1-yl-biphenyl-4-yl]-acrylic acid methyl ester (30 mg, 0.06 mmol) in 0.7 N NaOH was refluxed for 10 h. After removal of methanol, the residue was treated with cold water, acidified with 1M HCl, and the precipitate was filtered. Crystallization from isopropyl ether gave 10 mg (37%) of the pure product as white solid, m.p. 208 °C, *R*_f_ (petroleum ether: ethyl acetate 50:50) 0.15. ^1^H-NMR (300 MHz, DMSO-*d*_6_) δ: 7.74–7.61 (4H, m); 7.59 (1H, d, J = 16.5 Hz); 7.49 (1H, dd, J = 8.80, 2.11 Hz); 7.41 (1H, d, J = 2.11 Hz); 7.03 (1H, d, *J* = 8.8 Hz); 6.51 (1H, d, *J* = 16.5 Hz); 4.49 (1H, brs); 4.02 (2H, t, J = 6.4 Hz); 3.48 (2H, t, J = 6.38 Hz); 2.16–2.20 (9H, m); 1.90–1.78 (2H, m); 1.77–1.59 (8H, m); ^13^C-NMR (75 MHz, DMSO-*d*_6_) δ: 167.8, 157.8, 143.2, 142.1, 137.7, 132.5, 131.0, 128.7 (×2C), 126.5 (×2C), 126.3, 125.2, 119.2, 112.9, 67.6, 60.4, 3C missing due to the overlap with signal solvent, 36.6 (×4C), 29.4, 28.4 (×3C), 25.8. Anal. calcd. for C_29_H_34_O_4_: C,78.00; H, 7.67. Found: C, 78.11; H, 7.68.

*3-(3’-adamantan-1-yl)-4’-hydroxy-[1,1’-biphenyl]-4-yl)-2-cyanoacrylic acid* (**9**). A suspension of compound **22** [15] (150 mg, 0.34 mmol), methyl cyanoacetate (1 g, 10.1 mmol) and beta-alanine (119 mg, 1.34 mmol) in ethanol (26 mL) was heated at 50 °C for 4 h. The solvent was evaporated, and the yellow residue was dissolved in CH_2_Cl_2_ (25 mL). The solution was washed three times with water (35 mL), dried over anhydrous Na_2_SO_4_ and evaporated. The resulting solid was washed with ethanol to dissolve the starting cyanoacetate to give 151 mg (85%) of methyl 3-(3’-adamantan-1-yl)-4’-tetrbutyldimethylsilyloxy-[1,1’-biphenyl]-4-yl)-2-cyanoacrylate. m.p. 185 °C.

The above compound (100 mg, 0.19 mmol) was added to a solution (THF: H_2_O 1:1, 7.7 mL) of LiOH·H_2_O (40 mg, 0.95 mmol). The mixture was stirred 72 h at room temperature. THF was evaporated. The remaining aqueous phase was extracted with hexane, then acidified with 1N HCl to pH 2 and filtered to give 57 mg of a crude compound. Purification by reverse phase flash chromatography (CH_3_OH: H_2_O 3:1) gave 26 mg of the title compound. m.p 228 °C; ^1^H NMR (acetone-*d*_6_), δ: 1.78 (6H, s), 2.13 (3H, s), 2.17 (6H, s), 6.95–7.05 (1H, m), 7.40–7.50 (1H, m), 7.65–7.55 (1H, m), 7.77–7.90 (2H, m), 8.07–8.22 (2H, m),8.32 (1H, s), 10.06 (1H, s); ^13^C-NMR (75 MHz, DMSO-*d*_6_) δ: 169.2, 157.8, 156.5, 142.5, 135.94, 132.29, 129.74, 128.65 (×2C), 126.25 (×2C), 124.88, 124.70, 118.6, 116.02, 101.2, 3C missing due to the overlap with signal solvent, 36.63 (×3C), 36.34, 28.42 (×3C). Anal. calcd. for C_26_H_25_NO_3_: C,78.17; H, 6.31; N, 3.51. Found: C, 78.25; H, 6.32; N, 3.50.

*(E)-3-(3’-adamantan-1-yl)-4’-hydroxy-5’-((3-phenylureido)methyl)-[1,1’-biphenyl]-4-yl)acrylic acid* (**14**). To a suspension of compound **23** (33 mg, [14] in TEA (4 mL), DMF (2 mL) and DMSO (0.2 mL) were added, followed by phenyl isocyanate (30 µL). The mixture was stirred 5 days at room temperature. The solvent was evaporated and water (2 mL) was added, followed by 2N HCl (100 µL). The yellow solid formed was filtered and dried (80%). ^1^H NMR (300 MHz, DMSO-*d*_6_) δ: 10.05 (1H, s), 8.90 (1H, s), 8.05 (1H, m), 6.90–7.80 (12H, m), 6.50 (1H, d, J = 16.0 Hz), 4.31 (2H, d, J = 6.0 Hz), 2.21 (6H, s), 2.07 (3H, s), 1.80 (6H, s); ^13^C-NMR (75 MHz, DMSO-*d*_6_) δ: 168.5, 157.4, 154.6, 142.1, 139.6, 137.8, 132.7,130.0, 128.7 (×4C), 128.1, 127.9, 127.0, 126. 9, 126.3 (×2C), 124.2, 119.9, 118.5 (×2C), 46.8, 3C missing due to the overlap with signal solvent, 36.6 (×4C), 28.4 (×3C). Anal. calcd. for C_33_H_34_N_2_O_4_: C, 75.84; H, 6.56; N, 5.36 Found: C, 75.79; H, 6.55, N, 5.35.

### 3.2. NMR Investigation on the Interaction of Adarotene with a Model of Microorganism Membrane

Adarotene (4 mg) was dissolved in 0.6 mL of SDS (Sodium dodecyl sulfate) in D_2_O. The concentration of SDS solution (24.8 mM) was greater than critical micelle concentration (8.2 mM). The NMR spectra were carried out at 25 °C on a Bruker AV600 spectrometer operating at a frequency of 600.10 MHz, equipped with a z-gradient 5 mm TXI probe. Chemical shifts (ppm) were referenced to residual solvent signal at 4.78 ppm. The protons were assigned using an integrated series of 2D experiments such as ROESY, NOESY and TOCSY. ROESY spectra were recorded with spin-lock of 250 and 400 ms. Phase sensitive NOESY spectra were acquired in TPPI mode, with 2048 × 1024 complex FIDs. Mixing times ranged from 100 to 400 ms. TOCSY spectra were acquired with the use of a MLEV-17 spin-lock pulse (60 ms total duration). All spectra were transformed and weighted with a 90 ° shifted sine-bell squared function to 4K × 4K real data points.

### 3.3. Antimicrobial Activity

MICs and MBCs were determined according to CLSI (Clinical and Laboratory Standards Institute) guidelines, in triplicate [22,23]. Results are shown as modal values of the three replicates. According to MIC and MBC data, the killing rate was calculated as the ratio between MBC and MIC on which basis it is possible to define the compounds tested as bactericides (killing rate = 1–4), or bacteriostatic (killing rate > 4) [24].

Preparation of the solution containing molecules: for each molecule to be tested, solutions were made as DMSO stock, with a concentration of 2.560 mg/mL. Scalar dilutions were than prepared directly on plates by using Mueller Hinton Broth (MHB) as a growth medium (range 0.5–256 µg/mL).

Bacterial suspension preparation: cryostrains preserved at −80 °C were recovered by incubation in growth medium (Brain Heart Infusion Broth, BHI) for 24–48 h at 35 ± 2 °C followed by passages in agar growth medium (Columbia Blood Agar). Colonies with less than 30 h were suspended in MHB with the purpose to prepare a bacterial suspension with a turbidity of 0.5 MF, containing approximately 1.5 × 10^8^ cfu/mL. This suspension was than diluted in MHB until reaching a final concentration of 5 × 10^4^ CFU/well. Simultaneously, check in broth-microdilution only with solvent (in order to exclude that the effect of growth inhibition might be due to the drug solution solvent), positive growth control in absence of DMSO and drug, and negative purity control only with injection broth were performed.

Bacterial strains: activity of the synthesized compounds was determined by using standard reference strains and a collection of MDR clinical isolates:

*S. aureus* ATCC 25923 (reference strain): *mecA* negative; multi-susceptible.

*S. aureus* strain 1 (collection, MDR): *mecA* positive; MRSA, β-lactamase positive, resistant to ciprofloxacin, gentamicin and erythromycin.

*S. aureus* strain 2 (collection, MDR): *mecA* negative; β-lactamase positive, resistant to erythromycin, gentamicin, clindamycin and fusidic acid.

*E. faecalis* ATCC 51299 (reference strain): *vanB* positive.

*E. faecalis* strain 1 (collection, MDR) *vanA/B* negative; aminoglycosides high-level resistant to aminoglycosides (gentamicin and streptomycin) and ampicillin.

*E. faecalis* strain 2 (collection, MDR) *vanA/B* negative; aminoglycosides high-level resistant to aminoglycosides (gentamicin and strseptomycin) and ampicillin.

Presence of *mecA* and *VanA/B* genes was investigated by PCR as previously described [25,26]. β-lactamase production and high-level aminoglycosides resistance was determined according to CLSI guidelines. Bacterial isolates were defined as MDR according to definitions by Magiorakos et al. [17]. Susceptibility results were interpreted according to EUCAST criteria (Clinical breakpoints - bacteria v 10.0)

## 4. Conclusions

It has been recently found that CD-437 related synthetic retinoids exhibit antimicrobial activity on MRSA. Having in our hands a large number of retinoids containing the adarotene scaffold, we tested some representative derivatives as antimicrobial agents to have an insight of the most relevant structural features affecting the activity.

We found that the pattern of substitution on adarotene can be modulated to obtain selectivity for antibacterial over anticancer activity. For example, an analysis of the MICs of compounds **9**, **16** and **17** against *S*. *aureus* and *E. faecalis* (including some MDR strains) showed that they have a very good activity profile. In many cases, these compounds showed together with favorable MICs, a detectable bactericidal effect. Overall, the results showed that the shape and geometry of the molecules together with the presence of the key -OH and –COOH play a role on the antimicrobial activity. NMR experiments allowed to define the interaction between RRM and a model of microorganism membrane. The collected data confirmed that the scaffold of adarotene is promising for further developments of non-toxic antimicrobials active on resistant strains.

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
