# Peer review of "New Antimicrobials Based on the Adarotene Scaffold with Activity against Multi-Drug Resistant Staphylococcus aureus and Vancomycin-Resistant Enterococcus"

_antibiotics, 2021, doi:10.3390/antibiotics10020126_

Round 1
Reviewer 1 Report
- General comment
In the manuscript, it was found that adarotene-related synthetic retinoids showed a detectable bactericidal effect, together with favorable MICs. The manuscript represents the promising of adarotene-related molecules for utilization as non-toxic antimicrobials active on resistant strains.
2. Major revision
1) As the molecular weights of the compounds listed in Fig. 2 were slightly different each other, it is strongly recommended to explain and discuss the data using two numerical values, µM and µg/mL, in Table 1 and Table 2.
2) It is necessary to show the position of adamantyl moiety signal in Figure 4, in order to understand the sentence of line 220” The adamantyl group shows a contact with CH2-11 of SDS as well.”
3)In order to easily understand the sentences of line 226~227, it is strongly recommended to draw retinoid related molecules distributed in SDS micelles in Figure 5, referring to Figure 2d or Extended Data Figure 2 of Ref.12.
Author Response
General comment
In the manuscript, it was found that adarotene-related synthetic retinoids showed a detectable bactericidal effect, together with favorable MICs. The manuscript represents the promising of adarotene-related molecules for utilization as non-toxic antimicrobials active on resistant strains.
1) As the molecular weights of the compounds listed in Fig. 2 were slightly different each other, it is strongly recommended to explain and discuss the data using two numerical values, µM and µg/mL, in Table 1 and Table 2.
Answer: as suggested, we have calculated and added in Tables 1 and 2 the MIC and MBC of all the compounds expressed in microM concentration. We have added a sentence (lines 122-125) explaining that we have added MIC and MBC values in micromolar concentrations, since the molecular weight of the compounds are slightly different. We have reported that the trend of activities still remains the same.
2) It is necessary to show the position of adamantyl moiety signal in Figure 4, in order to understand the sentence of line 220” The adamantyl group shows a contact with CH2-11 of SDS as well.”
Answer: We thank the reviewer for the useful suggestion. We have modified figure 4 by introducing a new box with the detail of the intermolecular NOEs between SDS micelles and the adamantyl moety of adarotene. Accordingly, the caption of the figure has been modified.
3)In order to easily understand the sentences of line 226~227, it is strongly recommended to draw retinoid related molecules distributed in SDS micelles in Figure 5, referring to Figure 2d or Extended Data Figure 2 of Ref.12.
Answer: We mistakenly included the graphical abstract instead of the proper figure 5. We apologize for this mistake. We have now added the correct figure with a schematic representation of the NOEs observed in 2D ROESY spectrum of adarotene +SDS micelles and a drawing showing the position of adarotene in SDS micelles as internally oriented with respect to the SDS molecules.
Reviewer 2 Report
In the manuscript entitled “New antimicrobials based on the adarotene scaffold with activity against multi-drug resistant Staphylococcus aureus and vancomycin-resistant Enterococcus” the authors assessed the antibacterial activity of compounds with previously identified antitumor activity. The results are understandable and well described. I suggest only the following minor changes:
Figure 5 is invalid. I suppose the authors mistakenly included a graphic summary instead of the proper figure.
Please verify that the NMR spectra have been recorded on Bruker AMX 300 MHz (page 9) or Bruker AV600 spectrometer (page 10).
Author Response
In the manuscript entitled “New antimicrobials based on the adarotene scaffold with activity against multi-drug resistant Staphylococcus aureus and vancomycin-resistant Enterococcus” the authors assessed the antibacterial activity of compounds with previously identified antitumor activity. The results are understandable and well described. I suggest only the following minor changes:
Figure 5 is invalid. I suppose the authors mistakenly included a graphic summary instead of the proper figure.
Answer: The reviewer is right. We included the graphical abstract instead of the proper figure 5. We apologize for this mistake. We have now added the correct figure with a schematic representation of the NOEs observed in 2D ROESY spectrum of adarotene +SDS micelles and a drawing showing the position of adarotene in SDS micelles as internally oriented with respect to the SDS molecules.
Please verify that the NMR spectra have been recorded on Bruker AMX 300 MHz (page 9) or Bruker AV600 spectrometer (page 10).
Answer: the NMR spectra recorded to characterize the new compounds were performed using a Bruker AMX 300 MHz, whereas the experiments to investigated the attachment of adarotene to a model of microorganism membrane were performed using a Bruker AV600 spectrometer. We have better clarified this point in the text (lines 260, 319)
Reviewer 3 Report
The manuscript entitled New antimicrobials based on the adarotene scaffold with activity against multi-drug resistant Staphylococcus aureus and vancomycin resistant Enterococcus is pretty interesting and can be published in Antibiotics, however there are several issues that should be addressed prior to publication:
- 13C NMR spectra as well as elemental analysis (or MS) results should be given for novel compounds to confirm their structure.
- The results of DSD NMR investigation should be given and discussed in detail.
- Figure 5 does not correspond with its title. I believe that graphical abstract has been inserted instead of figure 5. Please correct.
- Under Table 1, there is an information about LogP calculation and there is no letter f in the Table.
Author Response
The manuscript entitled New antimicrobials based on the adarotene scaffold with activity against multi-drug resistant Staphylococcus aureus and vancomycin resistant Enterococcus is pretty interesting and can be published in Antibiotics, however there are several issues that should be addressed prior to publication:
- 13C NMR spectra as well as elemental analysis (or MS) results should be given for novel compounds to confirm their structure.
- Answer: as suggested, we have introduced 13C and elemental analysis results for all the new compounds tested (lines: 282-288; 304-307; 315-318)
- The results of DSD NMR investigation should be given and discussed in detail.
- Answer: We mistakenly included the graphical abstract instead of the proper figure 5. We apologize for this mistake. We have now added the correct figure with a schematic representation of the NOEs observed in 2D ROESY spectrum of adarotene +SDS micelles and a drawing showing the position of adarotene in SDS micelles as internally oriented with respect to the SDS molecules. The new figure better clarifies the results of the NMR investigation. We have also modified a sentence in the discussion (line 240)
- Figure 5 does not correspond with its title. I believe that graphical abstract has been inserted instead of figure 5. Please correct.
- Answer: We apologize for this mistake. We have now added the correct figure
- Under Table 1, there is an information about LogP calculation and there is no letter f in the Table.
- Answer: We have corrected the table caption.